# Perceptions of pandemic resume gaps: Survey experimental evidence from the United States

**Regina Bateson** [ORCID] *

Graduate School of Public and International Affairs, University of Ottawa, Ottawa, Ontario, Canada

* rbateson@uottawa.ca

**Data Availability Statement:** All data and replication files are publicly available from the Open Science Framework (https://osf.io/pqkbu/).

## Abstract

As a result of the COVID-19 pandemic, millions of people found themselves out of work in 2020 and 2021. Going forward, will their pandemic resume gaps be stigmatized or forgiven? In a recent survey experiment in the United States, I find that US adults have negative perceptions of individuals who were unemployed during the novel coronavirus pandemic. When asked to select among fictional applicants for a job opening in the hospitality industry, respondents prefer those who were employed continuously throughout the pandemic. Respondents are about 20% less likely to choose applicants with pandemic resume gaps, regardless of whether they were laid off, stopped working to supervise virtual school, or yo-yoed in and out of employment. Respondents also describe applicants with pandemic resume gaps in more negative terms, perceiving them as less hardworking, less dedicated, less professional, and less qualified than otherwise identical applicants who remained employed. Public opinion toward individuals with breaks in employment during the pandemic matters because it may affect public policy, and because stigma harms job seekers in multiple ways. Furthermore, the results of the experiment are consistent among survey respondents with hiring and managerial experience. While we should always be cautious about generalizing from survey experiments, these findings suggest that people who were out of work during the COVID-19 pandemic may face disadvantages when they return to the labor market.

## Introduction

Since the COVID-19 pandemic hit the United States in March 2020, an unprecedented number of Americans have found themselves out of work. Some were laid off, while others stopped working due to illness, concerns about health and safety, or new caregiving responsibilities. The unemployment rate spiked to 14.8% in April 2020 [1], a development so sudden and widespread that it had "no modern equivalent" [2]. By July 2021, 3.4 million Americans had been unemployed for 6 months or more [3]. At the same time, millions more Americans were not working, but they did not meet the criteria to be categorized as "unemployed" because they were not actively seeking new jobs [4]. These trends were mirrored around the world, as the employment rate collapsed in countries ranging from Britain [5] to Japan [6] to Mexico [7].

**Funding:** This study was funded by a small grant from (no grant number) from the Women, Gender, and Politics Research Section of the American Political Science Association, https://connect.apsanet.org/s16/. RB received the grant. The funders had no role in study design, data collection and analysis, decision to publish, or preparation of the manuscript.

**Competing interests:** The author has declared that no competing interests exist.

The effects of these pandemic-era job losses may linger for years, well after the acute phase of the coronavirus pandemic has ended. Long-term unemployment typically causes "unemployment scarring" [8], in addition to negative impacts on health and psychological wellbeing [6–17]. Even when job losses result from an exogenous shock like the Great Recession, some research finds that lengthy periods out of work can harm future employment prospects [18–21]–which may bode poorly for those who were out of work during the COVID-19 pandemic [22, 23].

Yet as one displaced worker explained, "it feels like you have permission to be unemployed [due to the pandemic]. Obviously, being let go wasn't based on my performance or ability to help the bottom line. It truly was a condition of nature, right?" [24] Media commentators have reinforced this impression, reassuring job seekers that pandemic resume gaps will not be held against them [25, 26]. And indeed, some studies find that spells of unemployment do not necessarily harm workers' prospects for re-employment [27–29].

So going forward, how will individuals who were out of work during the pandemic be perceived? Will their pandemic resume gaps be stigmatized or forgiven? In a recent survey experiment in the United States, I find preliminary evidence of stigma toward those who were unemployed during the COVID-19 pandemic. These attitudes exist among the general public, and among survey respondents with hiring or managerial experience.

Among the general public, individuals with pandemic resume gaps are viewed more negatively than those who remained employed in 2020 and 2021. When presented with a vignette experiment, a nationally representative sample of US adults prefers fictional job applicants who were employed continuously throughout the coronavirus pandemic, compared to those who were out of work during the first waves of the pandemic. The survey respondents also describe people with pandemic resume gaps less favorably. Consistent with signaling theory [30–33], fictional job applicants with long periods out of work or who yo-yoed in and out of employment are significantly less likely to be perceived as hardworking, dedicated, professional, or qualified.

These results are not entirely surprising; public opinion is often critical of the unemployed, as has been demonstrated across multiple countries [34–37]. However, public opinion toward people whose employment was interrupted by the COVID-19 pandemic is particularly important for several reasons. First, public opinion matters for public policy [38]. If public opinion is critical of those who were out of work during the COVID-19 pandemic, this may affect whether or not they are seen as deserving of government assistance during the economic recovery. In addition, social stigma has negative consequences for individuals who are out of work. Social networks, word-of-mouth, and personal connections—even with "weak-tie" acquaintances [39]—all matter in a job search [40–42]. If people with pandemic resume gaps are viewed adversely by their peers, they may miss out on informal interactions and recommendations that could lead to new job opportunities. Moreover, simply being aware of others' disdain ("stigma consciousness") can be harmful for the unemployed [43, 44].

Like the general public, survey respondents who have experience in hiring or supervisory roles are more likely to select fictional job applicants who were continuously employed throughout the pandemic. They also describe fictional applicants with pandemic resume gaps less positively than those who remained employed. While these results suggest the experiment's findings could plausibly apply to real-world hiring scenarios, it is important to note that this is not an audit study or a field experiment, so it does not offer direct evidence of hiring discrimination in the labor market. However, this study suggests that policymakers should be attuned to the potential for discrimination against those who were out of work during the COVID-19 pandemic. If real-world hiring managers and recruiters share the assessments of the survey respondents, individuals with gaps in employment during the pandemic may face unique challenges when they seek to rejoin the workforce.

## Methods and data

### Recruitment and respondents

From July 7–10, 2021, I conducted a pre-registered survey experiment with a nationally representative sample of 974 US adults. The survey data and pre-registration are available on the Open Science Framework (OSF), https://osf.io/pqkbu. Though not randomly drawn, the sample was constructed to match the US census on key demographics. The survey experiment was fielded using Lucid Theorem, a reputable source of online survey respondents [45].

Eligibility was restricted to individuals who consented and passed two attention check questions—a particularly important step given burgeoning concerns about inattentiveness among online survey-takers [46, 47]. 1,000 respondents began the survey, and 974 finished it. Partial data from 26 subjects who attritted was discarded. This study was approved by the University of Ottawa Research Ethics Board, file number S-05-21-6805. All human subjects provided written informed consent.

### Survey instrument and experimental setup

The survey began with questions about the respondents' demographic characteristics, professional backgrounds and work histories, political and social views, and experiences during the COVID-19 pandemic. Next, in the experimental module, respondents read two vignettes about different hiring scenarios (SI Table 1 in S1 File). The first vignette asked respondents to select a new server for a restaurant in their town or city. The second vignette asked them to select a new front desk clerk for a hotel.

Following each vignette, the respondents saw profiles of three fictional job applicants. The fictional profiles are summarized in Tables 1, 2 and presented in full in SI Tables 2, 3 in S1 File. Within each profile, several attributes were held constant, including age, parental status, work experience, education, and most recent wage. At the same time, in a research design reminiscent of a Goldberg paradigm experiment [48] or an audit study [49], race, gender, and pandemic-era work histories were randomly assigned. SI Tables 4–6 in S1 File summarize the distribution of the randomized characteristics.

The experiment included four possible pandemic employment trajectories: 1) continuously employed; 2) continuously unemployed; 3) yo-yo unemployment; and 4) supervised virtual

**Table 1. Characteristics of fictional applicants, first hiring scenario.**

|  | Applicant A | Applicant B | Applicant C |
|---|---|---|---|
| **Age** | 33 years | 28 years | 26 years |
| **Parental Status** | 3 children | 2 children | 2 children |
| **Education** | 1 year of college | High school graduate | 2 years of college |
| **Work Experience** | 12 years | 8 years | 4 years |
| **Most Recent Position** | Restaurant server | Bartender | Restaurant server |
| **Most Recent Wage** | $9 per hour | $11 per hour | $10 per hour |
| **Race** | *[Black / white]* | *[Black / white]* | *[Black / white]* |
| **Gender** | *[male / female]* | *[male / female]* | *[male / female]* |
| **Pandemic Employment History** | *[continuously employed / continuously unemployed / yo-yo unemployment / supervised virtual school]* | *[continuously employed / continuously unemployed / yo-yo unemployment / supervised virtual school]* | *[continuously employed / continuously unemployed / yo-yo unemployment / supervised virtual school]* |

Characteristics in italics were randomly assigned. This 2 x 2 x 4 randomization resulted in 16 different versions of each applicant profile. See SI Table 2 in S1 File for the full text of the applicant profiles in narrative form, as presented to the survey respondents.

**Table 2. Characteristics of fictional applicants, second hiring scenario.**

|  | Applicant D | Applicant E | Applicant F |
|---|---|---|---|
| **Age** | 32 years | 35 years | 29 years |
| **Parental Status** | 2 children | 3 children | 2 children |
| **Education** | High school graduate | 1 year of college | Bachelor's degree |
| **Work Experience** | 8 years | 11 years | 5 years |
| **Most Recent Position** | Hotel concierge | Hotel office assistant | Hotel clerk |
| **Most Recent Wage** | $17 per hour | $22 per hour | $20 per hour |
| **Race** | *[Black / white]* | *[Black / white]* | *[Black / white]* |
| **Gender** | *[male / female]* | *[male / female]* | *[male / female]* |
| **Pandemic Employment History** | *[continuously employed /continuously unemployed /yo-yo unemployment /supervised virtual school]* | *[continuously employed /continuously unemployed /yo-yo unemployment /supervised virtual school]* | *[continuously employed /continuously unemployed /yo-yo unemployment /supervised virtual school]* |

Characteristics in italics were randomly assigned. This 2 x 2 x 4 randomization resulted in 16 different versions of each applicant profile. See SI Table 3 in S1 File for the full text of the applicant profiles in narrative form, as presented to the survey respondents.

school. These conditions reflect the unusual range of workers' experiences during the COVID-19 pandemic. Due to high levels of COVID-19, lockdowns, or the changed business climate, some workers were laid off for long periods of time ("continuously unemployed"). Others cycled through a pattern of yo-yo unemployment: as restrictions and cases waxed and waned, they were repeatedly laid off, re-hired, and laid off again ("yo-yo unemployment"). In addition, the closure of schools for in-person classes prompted some parents and caregivers to reduce their work hours or stop working to supervise online learning for their children ("supervised virtual school").

After viewing the fictional applicants for each hiring scenario, respondents were asked to choose one applicant. The probability of being selected is the main dependent variable for this study. Furthermore, for three of the applicant profiles, respondents were asked to describe the applicants. Respondents were provided with a list of 11 adjectives, including both positive and negative attributes (SI Table 7 in S1 File). While viewing each applicant's profile, they were instructed to check all the terms that described the applicant, with multiple selections allowed. They also used a slider bar to indicate how much they would offer to pay the applicant, if they selected him or her. The attributes selected and hourly wages offered are used as additional dependent variables in the analysis.

## Results

The results of survey experiment are analyzed for three sets of respondents: the full sample, respondents with hiring or managerial experience, and additional subgroups of respondents. The full sample reflects US public opinion toward those who were out of work during the pandemic. Meanwhile, the restricted sample of respondents with hiring or managerial experience offers insights into external validity, and the additional sub-group analyses explore heterogenous treatment effects among respondents with different life experiences, political party affiliations, ideological beliefs, and attitudes toward COVID-19.

Because each of the fictional applicant profiles had slightly different characteristics—such as age, education, and years of experience—all models include fixed effects by applicant profile. This controls for differences across the applicant profiles, as well as any inadvertent ordering or labelling effects (the profiles were labelled A, B, C, D, E, F). In addition, all models use robust standard errors clustered by respondent.

## Public perceptions of pandemic resume gaps

Among the full sample of US adult respondents, fictional job applicants' pandemic work histories significantly affect their chances of being selected in the survey experiment. Individuals who worked throughout the pandemic are chosen most frequently (Fig 1). Fictional job applicants randomly assigned to the "continuously employed" condition are selected 38.5% of the time (95% confidence interval: 36, 41). By contrast, continuously unemployed applicants are selected 31.2% of the time (95% confidence interval: 28.8, 33.5), those who experienced yo-yo unemployment are selected 32% of the time (95% confidence interval: 29.6, 34.4), and those who stopped working to supervise virtual school are selected 31.5% of the time (95% confidence interval: 29.2, 33.9).

Put differently, having been out of work for a long period during the pandemic decreases an applicant's probability of being selected by about 20%. This result is similar regardless of the reason why the hypothetical job applicant was out of work. Applicants with a history of yo-yo unemployment do not seem to get extra credit for having briefly returned to work—perhaps because an erratic work history can send negative signals about job applicants' attitudes and "soft skills" [33]. And although workers who "opt out" to care for children usually face particular penalties [50], applicants who stopped working to supervise virtual school do not fare any worse than those who were laid off.

Contrary to expectations, the impact of a pandemic resume gap does not vary according to applicants' racial-gender identities, and this experiment does not find evidence of discrimination against Black and/or female job applicants. As noted in SI Table 8 in S1 File, respondents seem to prefer Black and female applicants. And in a null result, neither the fictional applicants' race, gender, nor employment history significantly affects the wages they are offered, perhaps due to strong anchoring effects. In the experimental vignettes, respondents were told

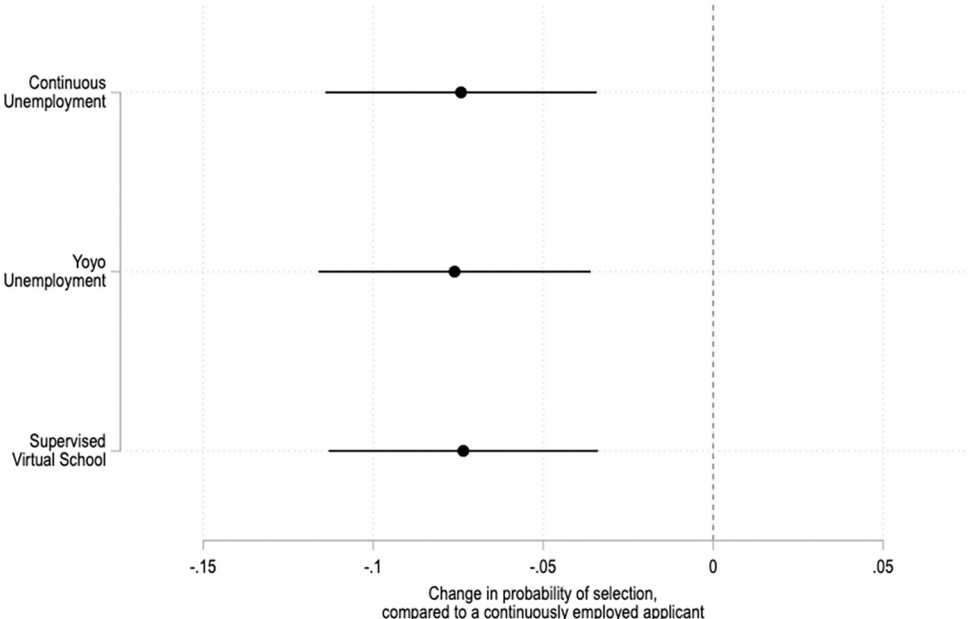

**Fig 1. How pandemic employment history affects the public's preferences.** Coefficients and 95% confidence intervals from an OLS regression with fixed effects by applicant profile and robust standard errors clustered by respondent. The reference category is a continuously employed applicant. The unit of analysis is the applicant profile. N = 5,844. Full results and robustness checks in SI Table 8 in S1 File.

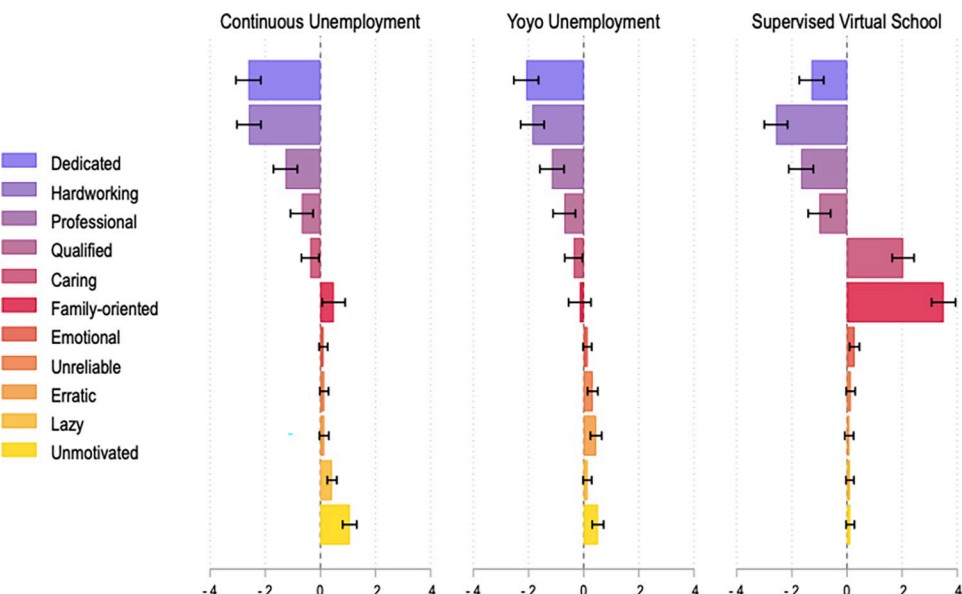

**Fig 2. Public perceptions of fictional job applicants, by pandemic employment history.** This figure reports coefficients from 11 different OLS regressions. The dependent variables are binary variables indicating whether each profile was described with a given adjective. All models include fixed effects by applicant profile and robust standard errors clustered by respondent. Black brackets are 95% confidence intervals. The unit of analysis is the applicant profile. For each regression, N = 2,922. Full results in SI Tables 9 and 10 in S1 File.

each applicant's most recent wage, and respondents tended to propose wages that hewed closely to those numbers.

However, as reported in Fig 2, pandemic-era employment history significantly affects the public's perceptions of job seekers. Survey respondents describe fictional job applicants who were continuously unemployed, experienced yo-yo unemployment, or supervised virtual school less positively than those who continued working; they are markedly less likely to be labelled hardworking, dedicated, professional, or qualified. These effects are large and statistically significant. For example, a continuously employed fictional job applicant is about 50% more likely to be called "hardworking," compared to someone who has been out of work since April 2020 (p<0.001). This is consistent with the idea that long periods of unemployment can signal low productivity [31, 32]. The results for the yo-yo unemployment condition are also consistent with prior research showing that frequent job-changing can be interpreted as a sign that an individual has a poor attitude toward work [33].

Respondents describe fictional job applicants who were out of work for different reasons in somewhat different terms (Fig 2). Consistent with prior research [51], continuous unemployment seems to signal a lack of motivation. Meanwhile, those who supervised virtual school are more likely to be seen as caring and family-oriented. These results are, in effect, a manipulation check; they show that respondents read, understood, and thought about the applicant profiles presented to them.

## Results for respondents with hiring or supervisory experience

In addition to providing insights into US public opinion, this study also offers preliminary data on how individuals with management and hiring experience perceive job applicants with pandemic resume gaps. Before they completed the experimental module, respondents were

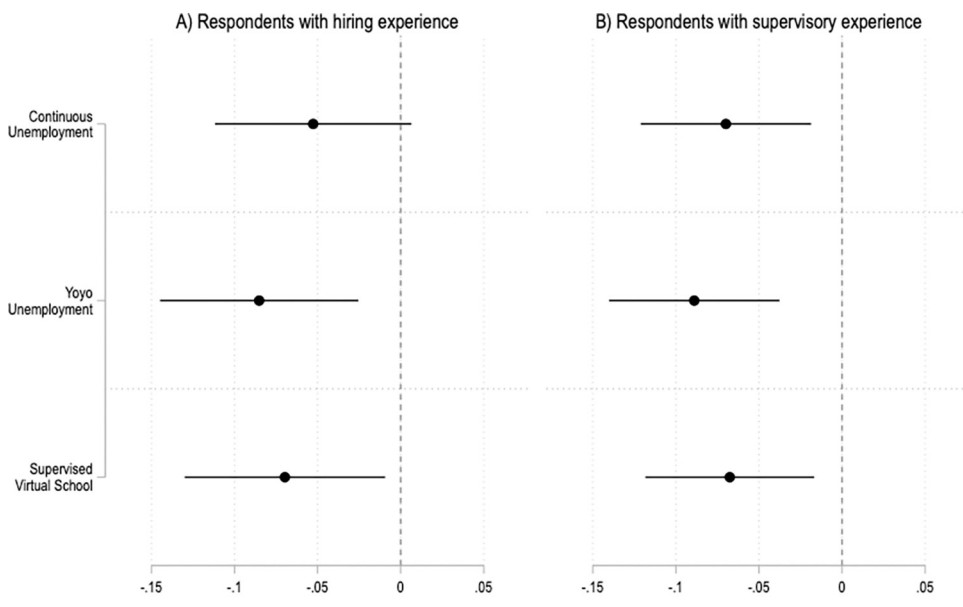

**Fig 3. Treatment effects among respondents with hiring and supervisory experience.** Coefficients and 95% confidence intervals from two OLS regressions with fixed effects by applicant profile and robust standard errors clustered by respondent. The reference category is a continuously employed applicant. The unit of analysis is the applicant profile. Panel A is based on data from respondents with hiring experience; N = 2,676. Panel B is based on data from respondents with supervisory experience; N = 3,600. Full results in SI Table 11 in S1 File.

asked whether they had ever participated in hiring an employee, and whether they had ever supervised anyone at work. 46% of respondents said they had hiring experience, and 62% said they had supervisory experience. The main findings of the experiment are largely consistent among these respondents (Fig 3), though the N is smaller, so the standard errors are larger than in the analysis with the full sample of respondents.

Among respondents with hiring experience, a history of yo-yo unemployment or supervising virtual school has a negative impact on the probability that a fictional job applicant will be selected to fill an open position (Fig 3, Panel A). These results are statistically significant at the p<0.01 and p<0.05 levels, respectively. Meanwhile, the impact of having been unemployed continuously is negative but only marginally significant, with p<0.1.

The results are similar among respondents with supervisory experience (Fig 3, Panel B), who are less likely to select fictional applicants with a history of continuous unemployment, yo-yo unemployment, or supervising virtual school during the pandemic, compared to those who remained employed. These results are all statistically significant at the p<0.01 level.

Like the general public, respondents with hiring and/or supervisory experience view applicants with pandemic resume gaps less positively than otherwise identical applicants who remained continuously employed throughout the pandemic (Fig 4). The direction, magnitude, and significance of these results is similar to those reported in Fig 2.

Overall, respondents with hiring and supervisory experience react to the experiment in much the same way as the general public: they display preferences for continuously employed job applicants, and they ascribe more negative characteristics to applicants with breaks in employment during the pandemic. While this suggests that pandemic resume gaps could plausibly affect hiring decisions in the labor market, we should be cautious about generalizing from this study. Survey experiments have strengths, in that they allow us to study emerging

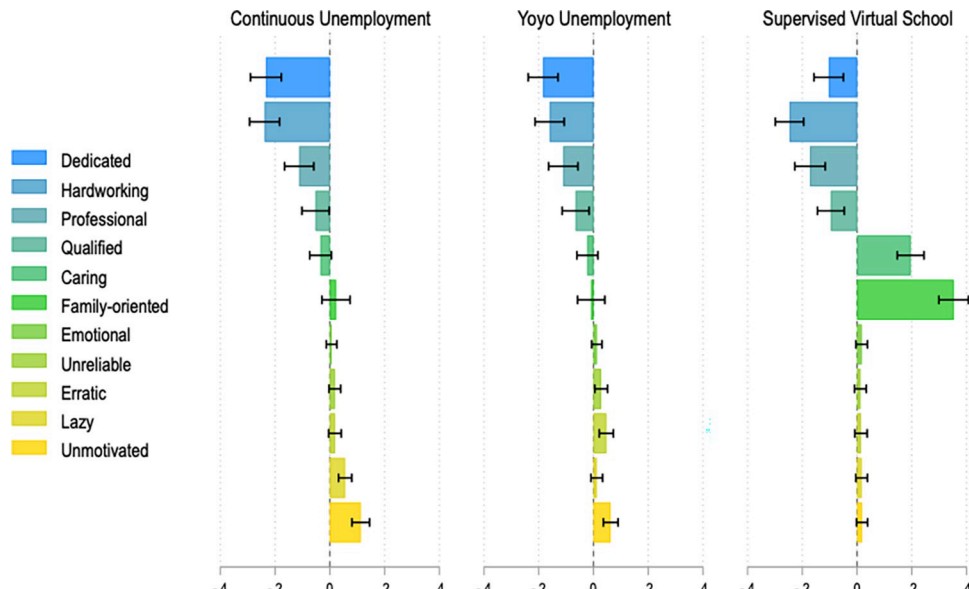

**Fig 4. Perceptions of fictional job applicants among respondents with hiring or supervisory experience, by pandemic employment history.** This figure reports coefficients from 11 different OLS regressions. The dependent variables are binary variables indicating whether each profile was described with a given adjective. All models include fixed effects by applicant profile and robust standard errors clustered by respondent. Black brackets are 95% confidence intervals. The unit of analysis is the applicant profile. For each regression, N = 1,932. Full results in SI Tables 12 and 13 in S1 File.

phenomena about which we may not yet have other sources of observational or field experimental data. Yet they also have limitations, particularly with regard to external validity. One study finds that paired vignette studies return results similar to real-world behavior [52], but the fact remains: survey experiments are necessarily simplified and decontextualized, so it is difficult to know whether people would react similarly in more complex, noisy situations.

Indeed, the results of this study are only partially congruent with audit studies examining how employment history affects callbacks for job interviews. While the yo-yo unemployment findings line up nicely with recent research on the negative signaling effects of frequent job changes [33], the relationship with audit studies on long-term unemployment is less clear—in part because the literature is so muddled, with inconsistent results. Some researchers find that recent long-term unemployment results in a lower rate of callbacks [21, 32, 53, 54], while others find no effect [27] or effects only under certain circumstances [28, 29, 55].

These differences may be due to contextual factors or design choices, which vary considerably across studies. Some studies use only male job applicants [21, 53], others use only female applicants [28, 29, 32]; some studies use young adult job applicants [21, 27, 32, 53–55], others use older adult applicants [28, 29]. Additionally, audit studies typically send fictitious applications to job openings for administrative assistants [28, 29, 32] and other white-collar positions in finance, banking, insurance, sales, and similar fields [27, 54]. This makes it difficult to draw direct comparisons with this survey experiment, which focuses on the hospitality industry. So while the present survey experiment provides initial insights into perceptions of pandemic resume gaps, further research will be needed to establish whether and how pandemic-era lapses in employment affect job seekers in the real world.

### Heterogenous treatment effects

To evaluate heterogenous treatment effects, the survey included questions about demographics, partisanship, ideology, and a range of experiences during the COVID-19 pandemic. Following best practices [56], these variables were measured pre-treatment, before respondents encountered the experimental hiring modules.

Surprisingly, having experienced the struggles described in the vignettes is not associated with greater tolerance for pandemic resume gaps. If anything, respondents who lost jobs and/or income due to COVID-19 may have slightly stronger preferences against job applicants who have been out of work during the pandemic (Fig 5).

The experimental results are also fairly consistent across respondents who see COVID-19 as a serious threat to public health and those who do not (Fig 6). However, there are notable differences by political party (Fig 7). In the United States, views on work and the welfare state shape political party identification, and vice versa [57, 58]. In addition, partisanship is strongly associated with attitudes and behaviors related to COVID-19 [59, 60]. And indeed, those relationships show up here. Compared to Democrats, respondents who identify as Republicans are more likely to penalize job applicants who were out of work during the COVID-19 pandemic (Fig 7). Ideology matters too, with moderate and conservative respondents driving the overall results of the experiment (Fig 8).

## Discussion

A recent survey experiment finds that in the United States, gaps in employment during the COVID-19 pandemic are perceived negatively by the general public and by individuals with hiring and managerial experience. These results are broadly consistent with Vishwanath's model of unemployment stigma in the labor market [31], but the relationship with audit

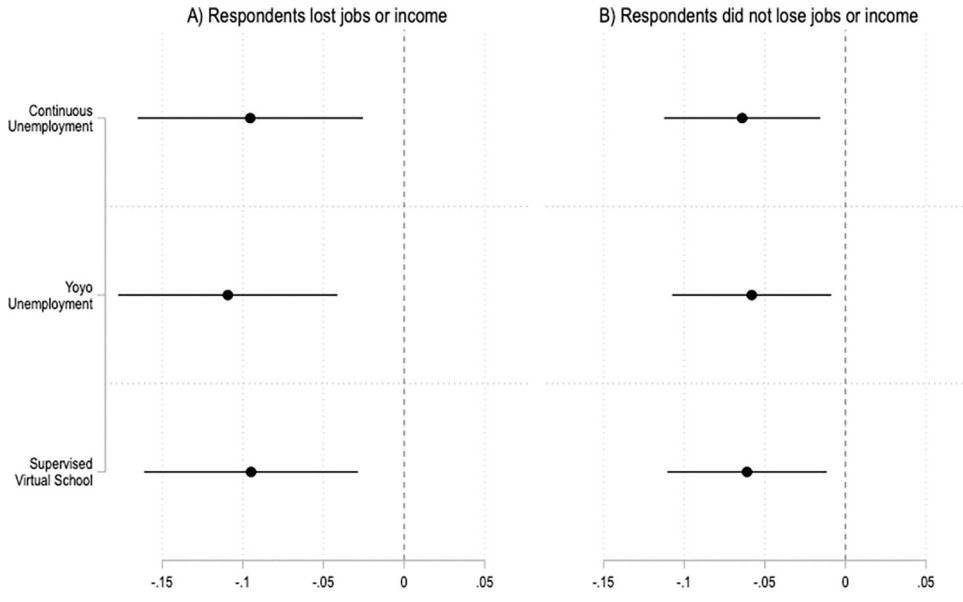

**Fig 5. Treatment effects by respondents' experiences during the COVID-19 pandemic.** Coefficients and 95% confidence intervals from two OLS regressions with fixed effects by applicant profile and robust standard errors clustered by respondent. The reference category is a continuously employed applicant. The unit of analysis is the applicant profile. Panel A includes data from respondents who lost jobs or income due to COVID-19; N = 2,088. Panel B includes data from respondents who did not lose jobs or income; N = 3,756. Full results in SI Table 14 in S1 File.

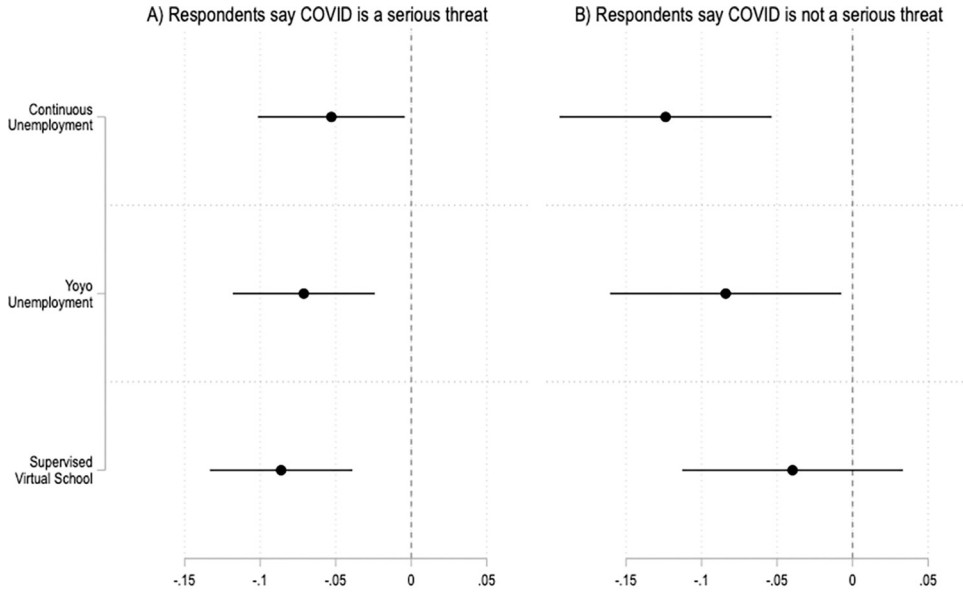

**Fig 6. Treatment effects by respondents' beliefs about the severity of COVID-19.** Coefficients and 95% confidence intervals from two OLS regressions with fixed effects by applicant profile and robust standard errors clustered by respondent. The reference category is a continuously employed applicant. The unit of analysis is the applicant profile. Panel A is based on data from respondents who say that COVID-19 is a serious threat; N = 4,062. Panel B is based on data from respondents who say COVID-19 is not a serious threat; N = 1,782. Full results in SI Table 15 in S1 File.

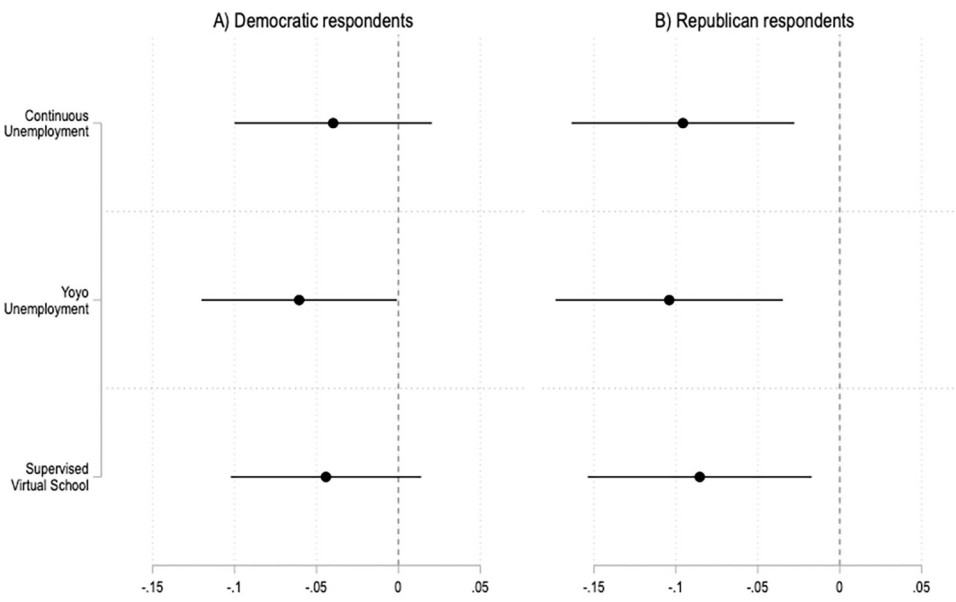

**Fig 7. Treatment effects by respondents' political party identifications.** Coefficients and 95% confidence intervals from two OLS regressions with fixed effects by applicant profile and robust standard errors clustered by respondent. The reference category is a continuously employed applicant. The unit of analysis is the applicant profile. Panel A is based on data from Democratic respondents; N = 2,556. Panel B is based on data from Republican respondents; N = 2,010. Full results in SI Table 16 in S1 File.

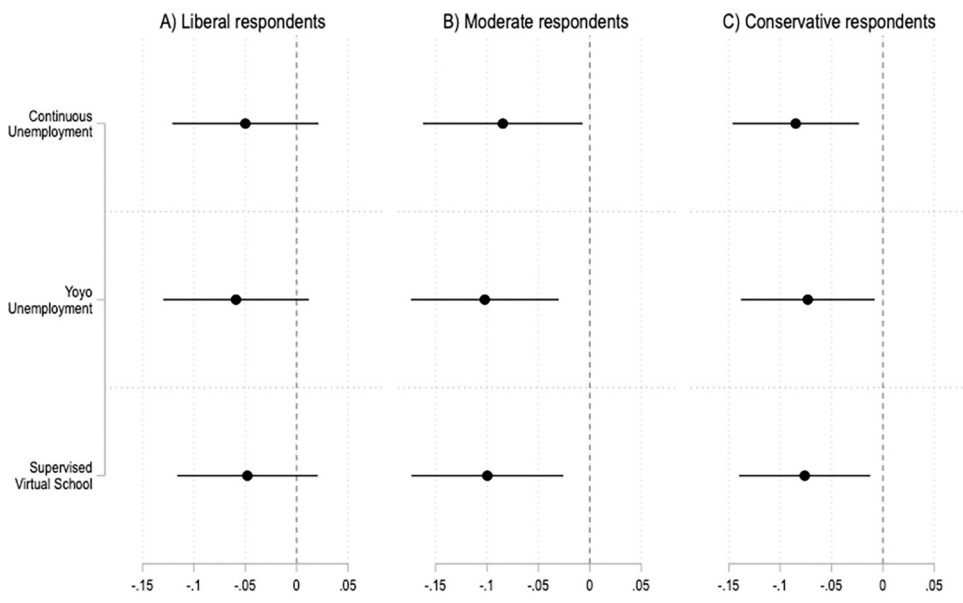

**Fig 8. Treatment effects by respondents' ideologies.** Coefficients and 95% confidence intervals from two OLS regressions with fixed effects by applicant profile and robust standard errors clustered by respondent. The reference category is a continuously employed applicant. The unit of analysis is the applicant profile. Panel A is based on data from liberal respondents; N = 1,782. Panel B is based on data from moderate respondents; N = 1,656. Panel C is based on data from conservative respondents; N = 2,406. Full results in SI Table 17 in S1 File.

studies on unemployment duration and callbacks from employers is less certain. Future research will be needed to thoroughly understand the real-world impact of pandemic resume gaps.

In this survey experiment, however, the results are clear: compared to fictitious job applicants who worked continuously throughout the novel coronavirus pandemic, individuals with breaks in employment during the pandemic are seen as less desirable hires. Survey respondents also perceive them as having less positive characteristics. All else being equal, a pandemic resume gap increases a fictional job applicant's risk of being seen as lacking in professionalism, qualifications, motivation, and dedication.

But do these results matter in the context of a sizzling job market? The impact of unemployment stigma is typically blunted in a tight labor market [28], and as vaccines have become available and public health restrictions have eased, US employers are scrambling to re-staff their operations. This is especially true in the hospitality industry, where many businesses have had difficulty recruiting enough workers to meet demand. Yet even so, the highest-quality jobs continue to receive large numbers of applications [61]. People who were out of work during the pandemic may be excluded from these plum opportunities, losing out on better wages, benefits, and working conditions.

Furthermore, the pace of hiring will presumably slow again at some point–and when that happens, job seekers with pandemic resume gaps could find themselves at a disadvantage. This prospect is particularly troubling because COVID-related unemployment has disproportionately affected people of color and women [1, 62–64]. So although this experiment does not find evidence of racial or gender discrimination per se, the stigma of a gap in employment during the COVID-19 pandemic could still result in setbacks for racial and gender equity in the United States.

## Supporting information

**S1 File.**
(DOCX)

## Acknowledgments

I am grateful for logistical assistance from the Telfer School of Management at the University of Ottawa, support from the Women, Gender, and Politics Research Section of the American Political Science Association, and feedback from Patrick Leblond and Kate Weisshaar.

## Author Contributions

**Conceptualization:** Regina Bateson.

**Data curation:** Regina Bateson.

**Funding acquisition:** Regina Bateson.

**Investigation:** Regina Bateson.

**Methodology:** Regina Bateson.

**Visualization:** Regina Bateson.

**Writing – original draft:** Regina Bateson.

**Writing – review & editing:** Regina Bateson.

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
