## [Decision Letter · Decision Letter 0]

11 Sep 2022

PONE-D-22-16366Unemployment scarring and the COVID-19 pandemic: How pandemic resume gaps affect perceptions of job-seekersPLOS ONE

Dear Dr. Bateson,

Thank you for submitting your manuscript to PLOS ONE. After careful consideration, we feel that it has merit but does not fully meet PLOS ONE’s publication criteria as it currently stands. Therefore, we invite you to submit a revised version of the manuscript that addresses the points raised during the review process.

One of the reviewers is negative and recommends rejection, mainly because of the not convincing external validity of the results.

The other reviewer is more positive and he/she acknowledges that the paper has some potential, but it still requires substantial work before it can be considered for publication.

After my careful reading of the paper, I agree with the second reviewer that your analysis is potentially interesting and the results may be policy relevant, but the current version of the paper suffers from a number of limitations and, as stated by both reviewers, it is not effective in explaining your theoretical arguments and the empirical results.

Although I believe it is a risky revision, I decided to give you the opportunity to thoroughly revise the paper following all the detailed comments of the second Reviewer.

I agree with both reviewers that the existing literature should be thoroughly reviewed, and more studies on other countries different from the US should be included, in order to be able to better interpret your results. Furthermore, I believe addressing the first main concern of reviewer 2 and, eventually use for the baseline estimates those with some experience in hiring, will further improve the paper.

Please let me remark that, given the two referee reports, the revision process will actually require to partly revise your empirical strategy and to rewrite substantially your manuscript.

We look forward to receiving your revised manuscript.

Kind regards,

Simona Lorena Comi

Academic Editor

PLOS ONE

Journal Requirements:

Reviewers' comments:

Reviewer's Responses to Questions

**Comments to the Author**

1. Is the manuscript technically sound, and do the data support the conclusions?

Reviewer #1: Partly

Reviewer #2: Yes

2. Has the statistical analysis been performed appropriately and rigorously? 

Reviewer #1: Yes

Reviewer #2: Yes

3. Have the authors made all data underlying the findings in their manuscript fully available?

Reviewer #1: No

Reviewer #2: Yes

4. Is the manuscript presented in an intelligible fashion and written in standard English?

Reviewer #1: Yes

Reviewer #2: Yes

5. Review Comments to the Author

Reviewer #1: The article deals with an important topic from the perspective of both the social sciences and society. It is well written and, although somewhat underdeveloped in its theoretical background as in the discussion on possible stratification consequences, still it provides clear arguments and comprehensible results, with interesting findings especially in the section dealing with heterogeneity at the group level. In the author's place, I would prefer to view the results as indicative of positive discrimination of those who are continuously employed, since no clear differences emerge between individuals with other labour market trajectories. As a further suggestion, I would try to use variables related to the age and health status of applicants, if available. That said, despite the robustness check provided by the authors, the potential external validity of the analytical design is not entirely convincing. Overall, regrettably I do not consider the manuscript, in its current form, suitable for publication in this journal.

Reviewer #2: Referee Report „Unemployment scarring and the COVID-19 pandemic: How pandemic resume gaps affect perceptions of job-seekers”

The paper provides a survey experiment on the signalling effect of resume gaps during the COVID-19 pandemic. The paper provides evidence for the negative effect of different resume gaps, indicating the potential negative consequences of the pandemic for future labor market outcomes. This original research tries to fill an important current gap in the literature on the negative consequences of unemployment during the pandemic.

I have two main concerns and one minor point for the current version of the manuscript. While they imply a major revision of the manuscript, given the great design and analysis, I have no doubt that the revision is feasible and would make a great future publication in PLOS ONE. This is why I recommend a revise and resubmit.

Main Concern 1:

The paper conducts a survey experiment on a national representative sample of the United States. However, the general population is not suitable to make implications about negative consequences during the hiring process. The paper provides mainly evidence for perceptions of job-seekers in the general population, not what is observable during the hiring process. While the perceptions of the general population are of importance for political interventions, a survey experiment on negative consequences in the labor market should focus on decision makers in the hiring process. I therefore welcome the robustness check which focuses on individuals with experience in the hiring process. Yet, the results focusing on this subgroup (Figure 7 Panel a)) provides no statistical significant evidence for negative consequences for continuous unemployment.

Suggestion:

I therefore recommend deciding to focus on individuals with hiring experience (which is common practice in such survey experiments) or to reframe the paper in perceptions of the general population. This is of importance, given the fact that the results are not in line with previous audit studies on the impact of the unemployment duration during the hiring process for the United States.

Main Concern 2:

My second main concern is related to the literature work of the manuscript, which does not cover a substantial literature relevant for the research question. In the following I give an overview of this literature and show why which papers are of relevance.

Motivated by the Great Recession, there is a growing literature of audit studies that look at the effect of the unemployment duration for the job finding rates. The advantage of these papers compared to survey experiments is that they allow to avoid demand effects. 3 out of 5 such audit studies do not find discrimination against long term unemployed (Farber et al., 2016; Farber et al., 2017; Nunley et al., 2017). The remaining two papers find evidence for negative effects during the hiring process, under specific circumstances (Ghayad, 2013; Kroft et al., 2013). Given that the findings stand in contrast with most of this literature, I would at least want to see a discussion that tries to explain these differences.

There are further paper for the European context, which might be of relevance for different reasons (Cohn et al., 2021; Eriksson and Rooth, 2014; Nüß, 2018; Oberholzer-Gee, 2018).

The paper needs a theoretical foundation why unemployment during the pandemic should effect labor market outcomes at al. Given the findings, the perceived lack of motivation indicates that the signalling theory is a good starting point, also supported by previous experiments (Oberholzer-Gee, 2008; Kroft et al., 2013).

The paper by Cohn et al. (2021) provides a laboratory experiment, a survey experiment as well as audit study and shows that yo-yo unemployment is interpreted as a signal for low reliability which leads to worsen labor market outcomes. This paper is closest to the manuscript and should be therefore considered. This might also help to bring the manuscript more in context of the literature.

Lastly, two papers consider business cycle effects of unemployment spells (Korft et al., 2013; Nüß, 2018). These experiments show that unemployment hast only negative effects on job findings in strong labor markets (i.e. low unemployment). In line with the signalling theory, unemployment spells are interpreted as a negative signall when unemployment is low and individuals should easily find a job. If people still do not find a job, employers get sceptical about the productivity and motivation of the applicants. In contrast, when unemployment is high and it is more difficult to find a job, employers interpret less into the unemployment. Given the fact that the results of the survey experiment find negative effects during the pandemic, independent of the reason for the unemployment spell, the manuscript should at least discuss its findings in context of this literature.

Minor Concern:

A last minor point is related to the title, which covers the term “Unemployment scar”. An unemployment scar, is the long run effect of long time past unemployment on current labor market outcomes. In contrast, negative consequences of recent/current unemployment is named an unemployment stigma (Vishwanath, 1989).

Suggestion:

So to analyse an unemployment scar in a survey experiment would mean to provide participants information about past unemployment. It can be ruled out that findings of survey experiments and audit studies are due to an unemployment scar because Eriksson and Rooth (2014) tested current and past unemployment experience in an audit study, providing evidence for discrimination based on current unemployment spells, while past unemployment spells had no effect. So technically calling it a scar is the wrong term and it would be great to be more precise with it.

References:

Cohn, A., Maréchal, M. A., Schneider, F., & Weber, R. A. (2021). Frequent job changes can signal poor work attitude and reduce employability. Journal of the European Economic Association, 19(1), 475-508.

Eriksson, S., & Rooth, D. O. (2014). Do employers use unemployment as a sorting criterion when hiring? Evidence from a field experiment. American economic review, 104(3), 1014-39.

Farber, H. S., Silverman, D., & Von Wachter, T. (2016). Determinants of callbacks to job applications: An audit study. American Economic Review, 106(5), 314-18.

Farber, H. S., Silverman, D., & Von Wachter, T. M. (2017). Factors determining callbacks to job applications by the unemployed: An audit study. RSF: The Russell Sage Foundation Journal of the Social Sciences, 3(3), 168-201.

Ghayad, R. (2013). The jobless trap. Northeastern University.

Kroft, K., Lange, F., & Notowidigdo, M. J. (2013). Duration dependence and labor market conditions: Evidence from a field experiment. The Quarterly Journal of Economics, 128(3), 1123-1167.

Nunley, J. M., Pugh, A., Romero, N., & Seals, R. A. (2017). The effects of unemployment and underemployment on employment opportunities: Results from a correspondence audit of the labor market for college graduates. Ilr review, 70(3), 642-669.

Nüß, P. (2018). Duration dependence as an unemployment stigma: Evidence from a field experiment in Germany (No. 2018-06). Economics Working Paper.

Oberholzer-Gee, F. (2008). Nonemployment stigma as rational herding: A field experiment. Journal of Economic Behavior & Organization, 65(1), 30-40.

Vishwanath, T. (1989). Job search, stigma effect, and escape rate from unemployment. Journal of Labor Economics, 7(4), 487-502.

6. PLOS authors have the option to publish the peer review history of their article (what does this mean?). If published, this will include your full peer review and any attached files.

Reviewer #1: No

Reviewer #2: No

---

## [Author Response · Author response to Decision Letter 0]

10 Nov 2022

Thank you for this opportunity to revise my manuscript, and thank you for the valuable feedback. I have made multiple significant changes to the manuscript, and I hope the new draft meets with your approval. 

While I appreciate the advice of both reviewers and the editor, I would particularly like to thank R2 for their exceptionally thorough and helpful review. R2's analysis and advice are spot-on, and I have implemented all the changes they recommended. I believe these efforts have significantly improved the manuscript, resulting in a more appropriate framing, better engagement with the literature, a stronger theoretical foundation, and more attention to studies from other countries. 

In line with the recommendations of the reviewers and the editor, I have made multiple major changes to the manuscript, including:

-reframing and rewriting the paper (including a new title!) 

-incorporating new literature, doubling the number of references in the manuscript, and adding studies and data from multiple other countries

-expanding, clarifying, and placing greater emphasis on the discussion of external validity 

Below, I describe these major changes, then I respond to additional points raised by each reviewer. 

Major Changes

1. Reframing and reorienting the manuscript

I agree with R2's observation that because the survey experiment was conducted with a nationally representative sample of US adults, it is primarily a study of public opinion toward those who were out of work during the COVID-19 pandemic. As R2 correctly points out, "the paper provides mainly evidence for perceptions of job-seekers in the general population." R2 therefore suggests that I either reframe the paper to focus exclusively on respondents with hiring experience, or to reframe the paper to be about "perceptions of the general population." 

In this revision, I have made major changes to emphasize that this is primarily a study of public attitudes toward the unemployed. I changed the title, rewrote the abstract, and edited the language throughout the paper to place greater emphasis on perceptions, rather than actual hiring of job applicants. For example, I have greatly reduced the use of the term "job-seeker." I also reduced the use of the terms "hiring" and "hiring decision" when discussing the survey experiment. Instead, in this draft, I describe how respondents selected, chose, or preferred fictional applicants with different employment histories. I believe this is more accurate and transparent, and this language is more consistent with what is actually happening in the survey experiment. 

In the new introduction, I situate the paper in relation to other studies of perceptions of the unemployed, and I articulate why public opinion toward the unemployed matters (see lines 93-105). I explain that public opinion shapes public policy, and stigma has multiple negative impacts on unemployed individuals. For instance, through "stigma consciousness" (Gurr and Jungbauer-Gans 2013), merely being aware of others' judgments can affect job search behavior and outcomes (Krug, Drasch, and Jungbauer-Gans 2019). While multiple prior studies have examined public perceptions of the unemployed in other contexts (Maassen and de Goede 1991, Eardley and Matheson 2000, Furåker and Blomsterberg 2003, Buffel and Van de Velde 2019), this is (to the best of my knowledge), to the best of my knowledge this manuscript is the first research to offer insights into perceptions of individuals who were out of work at the height of the COVID-19 pandemic. 

2. Expanded engagement with the literature & more theory

I would like to thank R2 for the extensive list of recommended references. I have cited all of R2's recommended references into this draft of the manuscript. In addition, I have added more than 20 other references to give the paper a better theoretical grounding, to make its contributions and limitations clearer, and to better situate it in an international context. 

Equipped with this additional research and reading, I have also revised the manuscript to clarify the theoretical interpretation of my results. I agree with R2 that this is really a study about unemployment stigma, not unemployment scarring (thank you, R2, for pointing that out!). I now discuss stigma at multiple points in the paper. I also argue that signaling is the most likely mechanism for the results of the experiment. The paper now includes citations to the literature on the negative signaling effects of periods of unemployment (Spence 1973, Vishwanath 1989) and frequent job changes (Cohn et al 2021). 

In addition, as recommended by R2, the paper now includes a substantial discussion of differences with prior audit studies on unemployment duration and callback rates for job interviews. This is primarily found on lines 291-308, though I also edited the introduction to clarify that the results in this literature mixed (see lines 70 and 77-78). 

Finally, addressing the editor's request for more discussion of studies from other countries, the revised manuscript includes new references to data and research from Japan, Britain, Germany, Switzerland, Sweden, Australia, the Netherlands, Mexico, and a multi-country study conducted across Europe. 

3. More emphasis on external validity

R1, R2, and the editor all raised concerns about external validity. I take that feedback seriously, and I have tried to reframe the manuscript to present the results more appropriately. 

In this revision, I have not used the results from respondents with hiring and managerial experience as the baseline models (although the editor recommended that I do so). I did not do this because doing so would contravene my pre-registered pre-analysis plan, and because it would be very unusual to present the results for a sub-group of respondents without first presenting the results for the full sample of respondents. 

However, I have taken taken multiple steps to expand and place greater emphasis on the results for the respondents with hiring and managerial experience. First, I moved this section up in the manuscript, so it appears more prominently at an earlier point in the manuscript. Second, I added a new figure (Figure 4) that shows how respondents with hiring and managerial experience describe the fictional job applicants. Third, the revised manuscript includes additional discussion of external validity on lines 280-290 and 305-308.

I have made best efforts to address R1's concerns about external validity, and I agree that we should always be careful about generalizing from survey experiments—as I note in the abstract (line 48) and multiple times in manuscript on (lines 284 and 372-74). However, despite their limitations, PLOS One regularly publishes survey experiments on a wide range of topics, including perceptions of COVID-19 policies (Zhang et al 2020), trust in government (Martin et al 2020), discrimination in the housing market (Ghekiere et al. 2022), choices among different COVID-19 vaccines (Kreps and Kriner 2022), pandemic-induced racial and ethnic prejudice (Kaushal, Lu, and Huang 2022), discrimination toward literary authors (Weinberg and Kapelner 2022), and evaluations of restaurants (Maezawa and Kawahara 2021). With the improvements in this draft, I hope this manuscript can eventually be published as well.

Additional Responses to Reviewers

R1

I would like to thank R1 for their attention to the manuscript. 

In this revision, I have adopted several of the changes recommended by R1. First, I have improved the theoretical background for the paper. Second, I have described the results as positive discrimination at several points (including lines 38-39 in the abstract and lines 86-87 in the body of the manuscript). Thank you for this helpful suggestion. 

I also appreciate R1's interest in analyzing the age and health status of the fictional job applicants, but I was not able to act on this request because age was held constant in the experimental vignettes and the profiles did not include any information about the fictional applicants' health status (as clarified in SI Tables 12-13). 

R2

I am deeply grateful for R2's comprehensive, constructive review, which was instrumental in informing my revisions. I also appreciate R2's praise for the manuscript's "great design and analysis." 

In response to R2's first main concern, I re-oriented most of the paper to focus more on public opinion toward individuals who were unemployed during the pandemic. However, I also partially adopted R2's second recommended strategy, which was to focus more on the results from respondents with hiring experience. That expanded discussion now appears more prominently in the manuscript, starting on line 233. 

In the expanded section on the results for respondents with hiring and managerial experience, I also address R2's concern about statistical significance of the coefficient on continuous unemployment. As noted in the manuscript, the standard errors are larger for that analysis, because the N is smaller (Lines 240-242). The p-value for that coefficient is 0.08, which is above the conventional 0.05 threshold but under 0.1. I acknowledge this in the manuscript and describe that result as only marginally statistically significant (Lines 257-258). If I were to combine the respondents with hiring experience together with the respondents with managerial experience (to create a larger N), this result would be statistically significant with p<0.05. However, I did not make this change in the manuscript because it would amount to p-hacking, which is not a good research practice. I hope R2 understands this rationale. 

In addition, I revised the title to remove the term "unemployment scarring," and I removed most discussions of "scarring" from the paper. I would like to thank R2 for pointing out this problem, and for advising me on the correct term ("unemployment stigma").

R2's literature references, the suggestion to consider signaling theory, and the recommendation to engage more with audit studies were all extremely helpful. Thank you so much! 

R2 went above and beyond in writing their review, contributing substantially to my revisions. Thank you again, I hope you enjoy reading this new version of the manuscript!

WORKS CITED

Buffel V, Van de Velde S. Comparing negative attitudes toward the unemployed across European countries in 2008 and 2016: The role of the unemployment rate and job insecurity. International Journal of Public Opinion Research. 2019;31(3):419-440.

Cohn A, Maréchal MA, Schneider F, Weber RA. Frequent job changes can signal poor work attitude and reduce employability. Journal of the European Economic Association. 2021;19(1):475-508.

Eardley T, Matheson G. Australian attitudes to unemployment and unemployed people. Australian Journal of Social Issues. 2000;35(3): 181-202.

Furåker B, Blomsterberg M. Attitudes towards the unemployed: An analysis of Swedish survey data. International Journal of Social Welfare. 2003;12:193-203. 

Ghekiere A, Verhaeghe P, Baert S, Derous E, Schelfhout S. Introducing a vignette experiment to study mechanisms of ethnic discrimination on the housing market. PLoS One 2022;17(10): e0276698. https://doi.org/10.1371/journal.pone.0276698

Gurr T, Jungbauer-Gans M. Stigma consciousness among the unemployed and prejudices against them: Development of two scales for the 7th wave of the panel study, "Labour Market and Social Security (PASS)." Journal for Labour Market Research. 2013;46:335-351.

Kaushal N, Lu Y, Huang X. Pandemic and prejudice: Results from a national survey experiment." PLOS One. 2022;17(4): e0265437. https://doi.org/10.1371/journal.pone.0265437

Kreps S, Kriner DL. Communication about vaccine efficacy and COVID-19 vaccine choice: Evidence from a survey experiment in the United States. PLoS One. 2022; 17(3): e0265011. https://doi.org/10.1371/journal.pone.0265011

Krug G, Drasch K, Jungbauer-Gans M. The social stigma of unemployment: Consequences of stigma consciousness on job search attitudes, behaviour and success. Journal for Labour Market Research. 2019;53(11):1-27. 

Maassen G, de Goede M. Changes in public opinion on the unemployed: The case of the Netherlands. International Journal of Public Opinion Research. 1991;3(2):182-194.

Maezawa T, Kawahara JI. A label indicating an old year of establishment improves evaluations of restaurants and shops serving traditional foods. PLoS One. 2021;16(11): e0259063. https://doi.org/10.1371/journal.pone.0259063

Martin A, Orr R, Peyton K, Faulkner N. Political Probity Increases Trust in Government: Evidence from Randomized Survey Experiments. PLoS One. 2020;15(2): e0225818. https://doi.org/10.1371/journal.pone.0225818

Spence M. Job market signaling. The Quarterly Journal of Economics 1973;87(3):355-374.

Vishwanath T. Job search, stigma effect, and escape rate from unemployment. Journal of Labor Economics. 1989;7(4):487-502.

Weinberg, DB, Kapelner A. 2022Do book consumers discriminate against Black, female, or young authors?" PLoS One. 2022;17(6): e0267537. https://doi.org/10.1371/journal.pone.0267537

Zhang B, Kreps S, McMurray N, McCain RM. Americans' perceptions of privacy and surveillance in the COVID-19 pandemic. PLOS One. 2020;15(12): e0242652. https://doi.org/10.1371/journal.pone.0242652

---

## [Decision Letter · Decision Letter 1]

8 Dec 2022

PONE-D-22-16366R1Perceptions of pandemic resume gaps: Survey experimental evidence from the United StatesPLOS ONE

Dear Dr. Bateson,

Thank you for submitting your manuscript to PLOS ONE. After careful consideration, we feel that it has merit but does not fully meet PLOS ONE’s publication criteria as it currently stands. Therefore, we invite you to submit a revised version of the manuscript that addresses the points raised during the review process.

Specifically,  I agree with Reviewer 2 that your paper is much improved and almost ready to be accepted for publication. Indeed,  Reviewer 2 suggested to accept the paper as it is. However, I would like to raise a couple of last very minor issues that, if taken into account, would support your empirical strategy even more.  

1) Your paper is missing a discussion (and a table) about the distribution of the characteristics of the applicants that you randomized (gender, race, age, number of children, level of education, previous wage, and so on…) across the applicants’ profiles you are using in the analysis (continuous unemployment, yo-yo unemployment, supervising school or continuous employment). Are they balanced?

Related to this, it would also be helpful to understand your analysis better if you could explain what you mean by “fixed effects by applicant profile” in the notes below each table and explain what variables you are controlling for. Are you already controlling for the applicant characteristics, the distribution of which I am asking you to document (gender, race, age, number of children, level of education, previous wage, and so on…)? If so, I encourage you to add a table with the full estimates in which you report the coefficients of these variables, even in SI.

2) It would be best if you numbered your Table in order of appearance.  The first Table mentioned in the paper is SI Table 11 (page 5, line 134). I suggest dividing the Tables provided in the SI into appendixes; in this way, Table 11 SI, together with all the tables about the survey (Table 12 and 13), could enter into Appendix A, and the table number could become Table A1 in Appendix A (Table A2 and A3). (very minor point)

We look forward to receiving your revised manuscript.

Kind regards,

Simona Lorena Comi

Academic Editor

PLOS ONE

Journal Requirements:

Reviewers' comments:

Reviewer's Responses to Questions

**Comments to the Author**

1. If the authors have adequately addressed your comments raised in a previous round of review and you feel that this manuscript is now acceptable for publication, you may indicate that here to bypass the “Comments to the Author” section, enter your conflict of interest statement in the “Confidential to Editor” section, and submit your "Accept" recommendation.

Reviewer #2: All comments have been addressed

2. Is the manuscript technically sound, and do the data support the conclusions?

Reviewer #2: Yes

3. Has the statistical analysis been performed appropriately and rigorously? 

Reviewer #2: Yes

4. Have the authors made all data underlying the findings in their manuscript fully available?

Reviewer #2: (No Response)

5. Is the manuscript presented in an intelligible fashion and written in standard English?

Reviewer #2: Yes

6. Review Comments to the Author

Reviewer #2: Thank you for this great resubmission. While I liked the design and analysis already in the first submission, The adjustments regarding the motivation and the theoretical bachground greatly imporved the paper.

Also thank you for not adjusting the analysis regarding the statistical sginfificance for participants with HR experience. In fact, signficance on the 10% level are more than acceptable. Making the significance level transparent was indeed the scietifically better solution.

7. PLOS authors have the option to publish the peer review history of their article (what does this mean?). If published, this will include your full peer review and any attached files.

Reviewer #2: No

---

## [Author Response · Author response to Decision Letter 1]

23 Dec 2022

To the editor and reviewers: 

Thank you for the invitation to revise and resubmit my manuscript. Once again, I would like to thank R2 for their detailed comments and constructive advice. Additionally, I appreciate the queries and suggestions from the editor. 

In this revision, I have made multiple changes to address the editor's comments and questions. 

There are 5 improvements in this revision:

1. Better explanation of the randomization in the survey experiment

The editor raised a number of questions about the randomization in the survey experiment. I agree that the randomization process was not thoroughly explained in the prior drafts; thank you for drawing my attention to this issue. 

For this revision, I created two new tables (Table 1 and Table 2 on pp. 6-7 of the manuscript). These tables summarize the characteristics that were held constant and and the characteristics that were randomized within each applicant profile. 

I hope Tables 1 and 2 clarify that there were only 3 characteristics randomized within the applicant profiles: pandemic employment history, race, and gender. I also added new text describing the randomization process (see lines 139-143 on p. 5). 

Additionally, in the captions for Tables 2 and 3, I explain that this is a 2 x 2 x 4 randomization, resulting in 16 different versions of each applicant profile.

I hope this additional information better explains what was held constant, and what was randomized. 

2. More information about the distribution of randomized characteristics

The editor also requested additional information about the distribution of the randomized characteristics across the applicant profiles.

In the SI, I have added 3 new tables summarizing the counts and percentages of the employment histories and racial and gender identities assigned to each fictional applicant profile. These are SI Tables 4-6, which are mentioned in lines 142-143 on p. 5 of the manuscript. 

The randomization was programmed using the "evenly present elements" function in Qualtrics, and overall it was quite well-balanced. However, as noted in SI Tables 4-6, within some applicant profiles, there are a few slight imbalances. I believe this may have resulted from the attrition of a small number of survey respondents, or from the variation inherent in the randomization process. 

However, even if we control for the race and gender assigned to each applicant profile, the impact of a pandemic resume gap remains negative and statistically significant (see SI Table 3, model 2). 

3. Additional robustness checks

As noted above, I have revised SI Table 8, which reports the main results from Fig 1 in the manuscript. 

Now SI Table 8 includes two new robustness checks (see models 2 and 3). Model 2 adds controls for the race and gender randomly assigned to each applicant profile, and Model 3 drops the fixed effects by applicant profile and the controls for race and gender. Across all these models, the coefficients on the pandemic employment histories are negative, statistically significant, and similar in magnitude. 

4. Better explanation of the fixed effects by applicant profile

To better explain the fixed effects by applicant profile, I have added new text on lines 188-191 of the manuscript (pp. 8-9). 

In short, Applicants A, B, C, D, E, and F each have slightly different biographies, with multiple characteristics that are fixed (not randomized) throughout the experiment, such as their age, number of children, level of education, years of work experience, most recent job title, and most recent wage. In addition, the applicants are labelled A, B, C, D, E, and F. These differences could make certain applicants seem more appealing, regardless of their employment histories. That is why I use fixed effects by applicant profile in my main models. The fixed effects by applicant profile control for the differences across the applicant profiles, allowing us to isolate the effect of the randomly assigned employment histories.

As recommended by the editor, I am now reporting the coefficients for the fixed effects by applicant in SI Table 8. I did not add the coefficients for the fixed effects to all the subsequent tables, because it would make all the tables much longer, and the coefficients for each applicant profile don't really convey any substantively interesting information. However, I am open to reporting the coefficients for the fixed effects in all tables if the editor thinks it is essential. 

5. Re-organizing and re-numbering the SI tables 

Finally, I have acted on the editor's request to re-number and re-organize the SI tables. There are now 17 tables in the SI, and they appear in the order that they are mentioned in the manuscript. 

I hope these changes meet with your satisfaction, and I look forward to hearing back from you soon. Thank you again for considering my work, and thank you for your detailed feedback and guidance.

---

## [Decision Letter · Decision Letter 2]

24 Jan 2023

Perceptions of pandemic resume gaps: Survey experimental evidence from the United States

PONE-D-22-16366R2

Dear Dr. Bateson,

We’re pleased to inform you that your manuscript has been judged scientifically suitable for publication and will be formally accepted for publication once it meets all outstanding technical requirements.

Kind regards,

Simona Lorena Comi

Academic Editor

PLOS ONE

Additional Editor Comments (optional):

Reviewers' comments:

Reviewer's Responses to Questions

**Comments to the Author**

1. If the authors have adequately addressed your comments raised in a previous round of review and you feel that this manuscript is now acceptable for publication, you may indicate that here to bypass the “Comments to the Author” section, enter your conflict of interest statement in the “Confidential to Editor” section, and submit your "Accept" recommendation.

Reviewer #2: All comments have been addressed

2. Is the manuscript technically sound, and do the data support the conclusions?

Reviewer #2: Yes

3. Has the statistical analysis been performed appropriately and rigorously? 

Reviewer #2: Yes

4. Have the authors made all data underlying the findings in their manuscript fully available?

Reviewer #2: Yes

5. Is the manuscript presented in an intelligible fashion and written in standard English?

Reviewer #2: Yes

6. Review Comments to the Author

Reviewer #2: (No Response)

7. PLOS authors have the option to publish the peer review history of their article (what does this mean?). If published, this will include your full peer review and any attached files.

Reviewer #2: No

---

## [Editor Report · Acceptance letter]

6 Mar 2023

PONE-D-22-16366R2 

Perceptions of pandemic resume gaps: Survey experimental evidence from the United States 

Dear Dr. Bateson:

I'm pleased to inform you that your manuscript has been deemed suitable for publication in PLOS ONE. Congratulations! Your manuscript is now with our production department. 

Kind regards, 

on behalf of

Professor Simona Lorena Comi 

Academic Editor

PLOS ONE